# Association between Retinol-Binding Protein 4 Levels and Preeclampsia: A Systematic Review and Meta-Analysis

**DOI:** 10.3390/nu14245201

**Published:** 2022-12-07

**Authors:** Hamdan Z. Hamdan, Tasneem Ali, Ishag Adam

**Affiliations:** 1Department of Basic Medical Sciences, Unaizah College of Medicine and Medical Sciences, Qassim University, Unaizah 51911, Saudi Arabia; 2Faculty of Medicine, Al-Neelain University, Khartoum P.O. Box 12702, Sudan; 3Faculty of Medicine, Sudan University for Science and Technology, Khartoum P.O. Box 407, Sudan; 4Department of Obstetrics and Gynecology, Unaizah College of Medicine and Medical Sciences, Qassim University, Unaizah 51911, Saudi Arabia

**Keywords:** adipokines, adipose-tissues, cytokines, preeclampsia, pregnancy-induced hypertension, retinol-binding protein 4, RBP4

## Abstract

Retinol-binding protein 4 (RBP4) is claimed to be associated with the development of preeclampsia, yet the reports are inconclusive. This systematic review and meta-analysis aimed to assess the association between RBP4 levels and preeclampsia. The PubMed, Google Scholar and ScienceDirect databases were searched for studies that investigated RBP4 levels in preeclampsia patients and compared them with normal controls. The meta-analysis was conducted by calculating the standardized mean difference (SMD) of RBP4 between cases and controls. The meta package with the R software was used to perform all statistical analysis. A total of 13 studies, comprising 569 cases and 1411 controls, met the inclusion criteria and were thus included in the meta-analysis. According to the random effect model, the SMD of RBP4 was significantly higher in women with preeclampsia compared with normal controls [SMD of RBP4: 0.55 ng/mL; 95% CI (0.06; 1.05); *p* = 0.028; I^2^ = 89%]. Likewise, the stratified meta-analysis showed the same pattern in the studies which measured RBP4 levels in the third trimester, as well as in the studies that investigated severe preeclampsia. Meta-regression did not identify any factor that significantly affected the overall estimate. There was no evidence of reporting bias (Egger’s test; t = 0.43; *p* = 0.587). This meta-analysis with high heterogeneity showed that higher levels of RBP4 were associated with preeclampsia risk. More longitudinal studies spanning the three trimester periods are needed to clarify the association of RBP4 and its dynamics in preeclampsia cases throughout pregnancy.

## 1. Introduction

Preeclampsia is a major medical disorder that frequently occurs during pregnancy. It is characterized by increased blood pressure and proteinuria in previously healthy women. It is a prevalent problem worldwide that affects around 4.5% of all pregnancies [1]. Without appropriate treatment, preeclampsia is associated with an increased risk of maternal and fetal deaths [2]. The definite cause of preeclampsia remains obscure. However, many predisposing factors, such as obesity, insulin resistance state, gestational diabetes, nulliparity and genetic factors are associated with the development of preeclampsia [3]. It has been reported that poor placental invasion is associated with ischemic reperfusion injury, which in turn generates ample amounts of free radicals that induce oxidative stress and release several angiogenic and inflammatory cytokines that are diffused into the maternal blood circulation [4]. These cytokines collectively initiate the endothelial changes that end up with preeclampsia [5].

Adipokines constitute a group of cytokines that are released from the adipose tissues. These adipokines are believed to affect the energy metabolism process, trigger an inflammatory response and enhance angiogenesis [6]. Retinol-binding protein 4 (RBP4) is an adipokine that is synthesized and released from the adipose tissues, just as liver cells are [7]. In addition to its major function as a transporter of Vitamin A and other retinoid derivatives in blood [7], it has been reported that it could affect the insulin hormone signaling cascade and mediate insulin resistance [8]. Moreover, an experimental animal model study has shown that mice with RBP4 overexpression have higher blood pressure levels compared with mice with normal RBP4 expression [9]. Taken together, all these premises have pointed to the involvement of RBP4 in the pathogenesis of preeclampsia.

Nevertheless, the published documents about the association between RBP4 and the development of preeclampsia in humans are inconclusive. Some studies have indicated increased levels of RBP4 in preeclampsia [10,11,12,13,14,15,16], while others have reported low levels [17] or no association [18,19,20,21,22,23]. Therefore, we conducted this systematic review and meta-analysis to assess the association between RBP4 and preeclampsia.

## 2. Materials and Methods

In this systematic review and meta-analysis, we followed the Preferred Reporting Items for Systematic Reviews and Meta-Analysis (PRISMA) 2020 recommendations [24], and the PRISMA checklist was added as a supplementary table (Appendix A).

### 2.1. Search Strategy and Sources

Two researchers (H.Z.H. and T.A.) independently searched three electronic databases (PubMed, Google Scholar and ScienceDirect) for relevant articles that investigated the association between RBP4 and preeclampsia since inception, until 3 November 2022. The detailed search strategy used for each database is available in Appendix A. To design the search strategy, we used the following [MeSH] terms: retinol-binding protein 4, adipokines, preeclampsia, pre-eclampsia, EPH complex and EPH gestosis, combined with the following non-MeSH terms: normotensive, normal pregnancy and healthy pregnant. The reference lists in the selected articles were further scanned for more eligible studies. Conflicts were resolved by discussing them with a referee reviewer (I.A.).

### 2.2. Eligibility Criteria

#### 2.2.1. Inclusion Criteria

We included articles if they (1) were published in peer-reviewed journals, reported serum/plasma levels of RBP4 in women with preeclampsia and were in the normal control group; (2) were written in the English language; (3) provided RBP4 concentration as the mean (standard deviation [SD]) or the median (interquartile range [IQR]) and reported the measuring unit; and (4) employed a case-control, cohort and cross-sectional study design.

#### 2.2.2. Exclusion Criteria

The articles were excluded if they (1) were clinical trials, experimental studies, review articles, letters to the editor, case series or conference abstracts; (2) reported RBP4 without the measuring unit or without related measures of dispersion, namely SD, range or 25th–75th IQR; (3) measured RBP4 in patients with preeclampsia and another chronic disease or another pathology; and (4) were written in a language other than English.

### 2.3. Data Extraction

Two researchers (T.A. and H.Z.H.) independently extracted the studies’ data using a structured sheet, guided by the Joanna Briggs Institute reviewers’ data extraction form [25]. The extracted data included the name of the first author, publication year, study’s location, continent, design and sample size (number of cases/number of controls), levels and assay method for RBP4, type of collected sample, parity status, trimester at which RBP4 was measured and maternal age and preconception body mass index (BMI) if available.

### 2.4. Definition of the Primary Outcome

In this study, the primary outcome is the association between the plasma/serum levels of RBP4 in pregnant women with preeclampsia compared with normotensive pregnant controls. Preeclampsia is defined according to the following criteria used by the American College of Obstetricians and Gynecologists [26]: detection of systolic blood pressure ≥ 140 mmHg and diastolic blood pressure ≥ 90 mmHg, as well as the presence of proteinuria > 300 mg/day.

### 2.5. Study Quality Assessment

Two researchers (T.A. and H.Z.H.) used the Newcastle–Ottawa Scale (NOS) to assess the methodological quality of each included study. The NOS assesses each study based on three major principles: recruitment of the participants, comparability of the study group and ascertainment of the outcomes of interest. Based on the NOS, the study is considered of high quality if it scores ≥ 7 stars. The maximum score in the NOS is 9 stars [27].

### 2.6. Statistical Analysis

All the data in this meta-analysis were analyzed by using the meta package [28] in the open-source statistical software R. The standardized mean difference (SMD) of RBP4 between the cases and the controls was calculated after applying the function “metacont”. In brief, the mean of RBP4 in a case group is subtracted from that in a control group and then divided by the difference of standard deviation between the case and control group. The pooled difference in mean for all studies is standardized by the pooled SD differences and corrected for bias according to modified Hedge’s methods [29]. We used Cochran’s Q with *p* < 0.010 and *I*^2^ > 50% as an indicator of significant inter-study heterogeneity [30,31]. If the meta-analysis indicated low heterogeneity, then the fixed effect model was followed; otherwise, the random effect model was used. Sensitivity analysis was conducted to identify studies with great impact and significant heterogeneity after their deletion. The presence of reporting bias was assessed subjectively by depicting studies in the funnel plot and quantitatively by performing an Egger’s test. Stratified meta-analysis was conducted to identify the possible sources of inter-study heterogeneity. Accordingly, the studies were grouped based on the timing of measuring RBP4 (first versus third trimester), studies that investigated the severity of the disease (severe versus mild to moderate preeclampsia), study’s continent (Asia versus Europe and America) and assay method for RBP4 (ELISA versus other methods). Meta-regression analysis was conducted to recognize possible factors (study’s publication year, location, total sample size, etc.) that significantly affected the SMD of RBP4. The Kappa statistics were used to evaluate the agreement between the researchers [32].

## 3. Results

### 3.1. Study Selection and Characteristics

The detailed search in the databases identified 263 articles. Of these, and after removing the duplicates and animal model studies, 62 articles remained eligible for screening. In the screening, based on their titles and abstracts, 20 articles were further excluded since they presented studies of other adipokines. After the retrieval of the articles, 29 were excluded further: 16 studies investigated other adipokines, 11 were found to be irrelevant, one did not mention the measuring unit of RBP4 and one examined preeclampsia in already diabetic patients. Finally, 13 articles were included in this study (see Figure 1). The researchers had high inter-rater agreement on study inclusion (Kappa = 0.86, *p* < 0.001). All included studies were of intermediate to high quality; their features are detailed in Table 1.

### 3.2. Association between RBP4 and Preeclampsia

Thirteen observational studies were included in this meta-analysis [10,11,12,13,14,15,16,17,18,19,21,22,23]. These studies enrolled a total of 569 preeclamptic women and compared them with 1411 pregnant controls. The pooled SMD of RBP4 showed significantly higher levels of RBP4 among preeclamptic cases than in normal pregnant controls [SMD = 0.55 ng/mL; 95% CI (0.058; 1.050); *p* = 0.028; random effect model] (Figure 2). Heterogeneity measures (Cochrane Q = 108.83; *p* = <0.0001) and (*I*^2^ = 89%) both indicated the presence of significant inter-study heterogeneity (Figure 2).

Sensitivity analysis was performed to recognize any study that significantly changed the overall effect upon deletion. None of the studies affected the heterogeneity measures significantly after deletion; therefore, none were excluded (Appendix A).

We investigated the presence of reporting bias visually, by depicting the studies in the funnel plot, and quantitatively, by performing an Egger’s test. The funnel plot showed a symmetrical plot of the included studies (Figure 3) and the result of the Egger’s test (*t* = 0.43; *p* = 0.587) was found nonsignificant.

### 3.3. Stratified Meta-Analysis and Meta-Regression

We performed a stratified meta-analysis of the included studies, based on the trimester at which RBP4 was measured: third trimester [10,11,12,13,14,15,17,18,22,23] and first trimester [16,19,21]. Additionally, the studies were divided based on those that examined only severe preeclampsia cases [12,15,16,23] and those that investigated the whole disease spectrum (severe and mild to moderate preeclampsia) [10,11,13,14,17,18,19,21,22]. The stratification was also performed based on each study’s continent: Asia [11,12,13,15,17,22,23] and Europe and America [10,14,16,18,19,21]. The assay method for RBP4 was also used to divide the studies into the ELISA group [10,11,13,14,15,17,18,19,21,22,23] and the group that used other methods [12,16].

The findings of the stratified meta-analysis are summarized in Table 2. The studies that measured RBP4 in the third trimester had higher levels of RBP4 in the cases compared with the controls [SMD = 0.70; 95% CI (0.08; 1.331); *I*^2^ = 90%]. A similar pattern was observed in the studies that investigated severe preeclampsia [SMD = 0.71; 95% CI (0.27; 1.15); *I*^2^ = 61%].

Although we noticed that the RBP4 SMD level increased by the years of publication, which can be attributed to the increased sensitivity of ELISA used, meta-regression analysis did not show any factor (sample size, continent and design, trimester of RBP4 measurements) that significantly changed the SMD of RBP4, including the study’s publication year (Table 3).

## 4. Discussion

The major finding in this meta-analysis of 13 observational studies is that the RBP4 levels are higher in patients with preeclampsia compared with normal pregnant controls. The same pattern is observed in the studies that measured RBP4 during the third trimester of pregnancy and in studies that investigated severe preeclampsia only. To the best of our knowledge, this is the first meta-analysis that investigated the association between RBP4 levels and the risk of preeclampsia. However, two previously published meta-analyses on the association between RBP4 and the risk of gestational diabetes mellitus (GDM) have also reported higher levels of RBP4 among GDM cases [33,34]. Notably, both preeclampsia and GDM usually develop in previously healthy women, whose symptoms usually commence during the second trimester of their pregnancy, with some variations. Moreover, preeclampsia and GDM are both well associated with the insulin resistance state [35,36]. Recently, RBP4 has been identified as a leading factor for endothelial dysfunction, a landmark pathology observed in preeclampsia as well as in GDM [37,38]. Perhaps RBP4 is the key modulator for the pathogenesis of both preeclampsia and GDM.

The stratified meta-analysis indicates that RBP4 levels are significantly higher in the third trimester of pregnancy compared with the first trimester. This finding should be taken with caution, as only three studies in this meta-analysis have evaluated the RBP4 levels in the first trimester of pregnancy [16,19,21]. Of these, two have reported insignificant differences [19,21] and only one has found significantly higher RBP4 levels [16]. From this paucity of studies, it is inconclusive whether first-trimester levels of RBP4 are associated with the development of preeclampsia. It is worth mentioning that RBP4 is a well-associated marker of insulin resistance [39]. Indices of insulin resistance are highly correlated with the gestational age in normal pregnancy as well as in preeclampsia [13,40]. However, RBP4 is not correlated with the gestational age in normal pregnancy [41], which may exclude the involvement of adipose tissue as a source of the increment of RBP4 during preeclampsia [14]. In this meta-analysis, nine of the 13 studies have recruited controls matched for BMI. Therefore, some scientists have proposed that liver dysfunction observed in preeclampsia is perhaps the major source of this increment in RBP4 [42]; renal dysfunction also complicates preeclampsia and may result in the excretion of less RBP4 in the urine [43]. Interestingly, RBP4 resists insulin action, particularly in blood vessels, which in turn curtails the nitric-oxide (NO) mediated vasodilatory action [44]. This may precipitate further blood vessel dysfunction observed in preeclampsia [45].

In this study, the stratified meta-analysis, based on preeclampsia severity, also indicates that RBP4 is significantly higher in the group with severe preeclampsia only. It has been mentioned that RBP4 is significantly higher in cases that develop severe preeclampsia compared with mild cases [13]. Moreover, Yliniemi et al. have found significantly higher levels of RBP4 in early-onset severe preeclampsia compared with controls [16]. Likewise, Inoue et al. have reported significantly increased levels of RBP4 in cases of severe pregnancy-induced hypertension compared with normal controls. Although insulin and Homeostatic Model Assessment of Insulin Resistance (HOMA-IR) levels are insignificantly higher in the cases than in the controls, the blood glucose levels are significantly higher in the pregnancy induced hypertension (PIH) group compared with the normotensive control group [12]. Moreover, RBP4 is found to be positively correlated with systolic blood pressure in severe preeclampsia [23]. Perhaps the severity of preeclampsia can be attributed to the insulin resistance that is mediated by high levels of RBP4.

Nulliparity is a known risk factor for developing preeclampsia. Nulliparous women have a 1.5 to 5.5 times increased risk of preeclampsia compared with multiparous women [46]. Maternal immune reaction to paternal sperm antigen is among the hypotheses that have been postulated to explain the mechanism by which nulliparity contributes to the pathogenicity of preeclampsia [47]. Since RBP4 is considered a cytokine [6] that can be an inflammatory mediator involved in the pathogenicity of preeclampsia, we hypothesized that its level is expected to be increased among nulliparous women in comparison to multiparous. However, only 4/13 studies [13,14,19,21] have recruited nulliparous women in their studies and one [13] of these four studies presented an analyzable RBP4 for nulliparous women. This limited data hindered us from performing a stratified meta-analysis based on the parity status, and hence, hindered us from drawing a reasonable judgement on the effect of parity on RBP4 levels.

Although the included studies in this meta-analysis are of moderate to high methodological quality and no evidence of reporting bias is detected, inter-study heterogeneity remains high. Zhang et al. [23] and Vaisbuch et al. [14] have both included cases of a mixture of early- and late-onset preeclampsia patients, while Yliniemi et al. [16] have only included patients with early-onset severe preeclampsia (<32 weeks of gestation). We think that these studies collectively contribute significantly to the overall inter-study heterogeneity, yet the sensitivity analysis indicates that no individual study has significantly changed the heterogeneity measures.

In this study, we observed that the reported levels of RBP4 are widely different between the included studies. This can be attributed to the differences in genetic factors that can affect the expression of RBP4 and hence, its levels [48]. Moreover, the degree of obesity is a documented, determining factor for the levels of RBP4 [7]. Furthermore, different assay methods were used in the included studies to assess the levels of RBP4, such as ELISA, immunoassay and Dot-plot. There is a notable variation in the sensitivity of these different methods [49]. Even the ELISA, which is a widely used method, has a different sensitivity based on the affinity of the antibody to RBP4 [49]. Taken together, these factors can contribute to the inter-study variations of the levels of RBP4 observed in this study.

The current meta-analysis has some limitations that need to be addressed for a better interpretation of the results. First, in the stratified meta-analysis, some of the groups only have three studies (stratification based on the trimester of pregnancy). This small number of studies with a small sample size will decrease the power for the estimation of RBP4 and increase the risk of bias. Moreover, only four studies recruited nulliparous women; however, they did not present their RBP4 separately. Accordingly, we could not perform stratified meta-analysis according to parity status, hence, we deduced a meaningful conclusion regarding the parity effect on RBP4 levels. Second, most of the included studies are retrospective case-control studies, so causality cannot be deduced. Third, inter-study heterogeneity is high, and the stratified meta-analysis and meta-regression could not identify major sources of this heterogeneity. Fourth, although RBP4 and preeclampsia both pertain to insulin resistance, a limited number of studies have investigated this relationship, so we could not include it in the meta-analysis. Therefore, further research is needed, which should use a larger sample size, a prospective design and longitudinal sampling for RBP4 to provide better insights into the RBP4 dynamics in preeclampsia cases.

## Figures and Tables

**Figure 1 nutrients-14-05201-f001:**
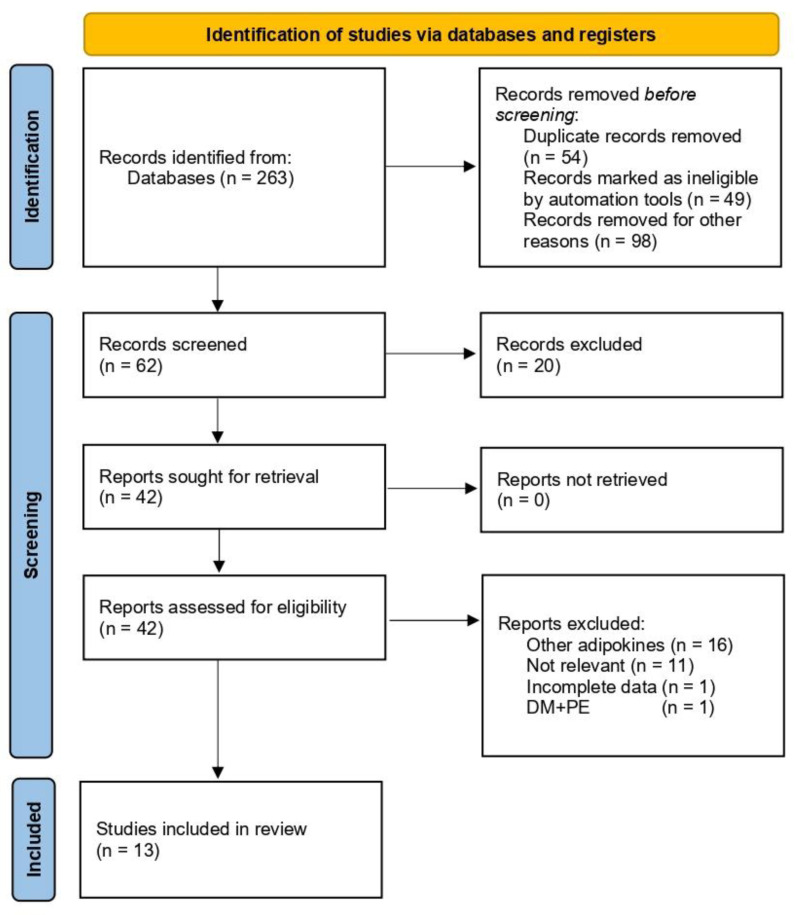
Flow chart showing the study selection process.

**Figure 2 nutrients-14-05201-f002:**
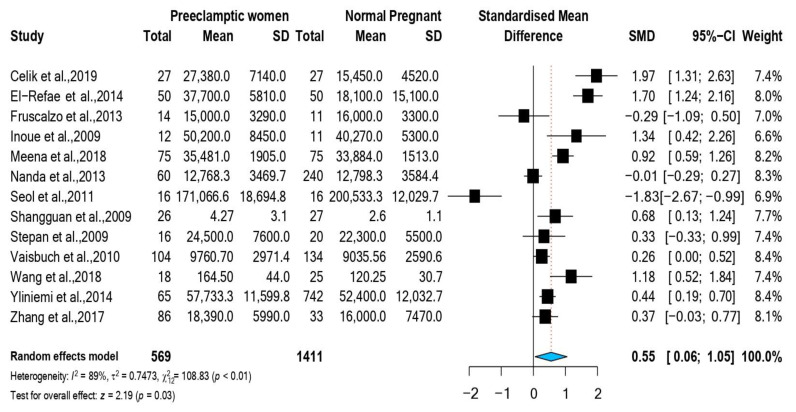
Forest plot for the overall meta-analysis [10,11,12,13,14,15,16,17,18,19,21,22,23].

**Figure 3 nutrients-14-05201-f003:**
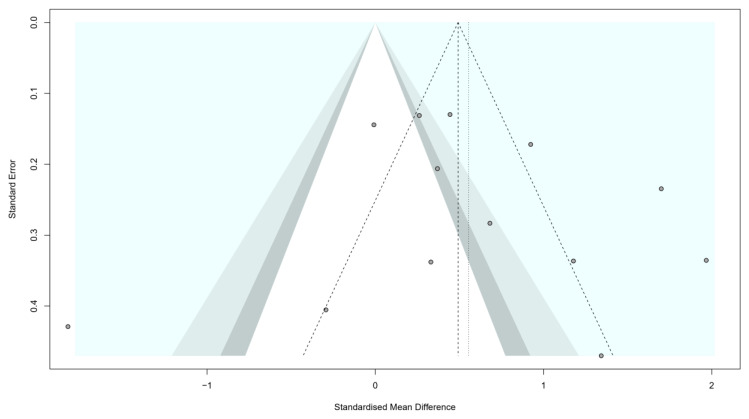
Funnel plot for reporting bias assessment.

**Table 1 nutrients-14-05201-t001:** Study characteristics included in the meta-analysis.

1st Author, Year, Citation	Study Design	Country	NOS Score	No. Preeclampsia/ No. Controls	RBP4 Mean (SD) Preeclampsia ng/mL	RBP4 Mean (SD) Controls ng/mL	Maternal Age Cases Mean (SD): Controls Mean (SD):	Maternal BMI Cases: Mean (SD) Controls: Mean (SD)	%Nulliparous among Cases/ %Nulliparous among Controls	Assay
Celik et al., 2019 [10]	Case-Control	Turkey	5	27/27	27,380 (7140)	15,450 (4520)	Cases: 30.59 ± 5.73 Controls: 29 ± 5.96	Cases: 28.76 ± 2.12 Controls: 27.97 ± 2.67	-	ELISA
El-Refae et al., 2014 [11]	Case-Control	KSA	5	50/50	37,700 (5810)	18,100 (15,100)	Cases: 33.25 ± 6.1 Controls: 31.65 ± 5.47	Cases: 28.28 ± 1.4 Controls: 27.06 ± 2.04	-	ELISA
Fruscalzo et al., 2013 [21]	Case-Control	Italy	8	14/11	15,100 (3290)	16,100 (3300)	Cases: 34.93 ± 5.31 Controls: 32.45 ± 5.56	Cases: 28 ± 7.4 Controls: 24 ± 2.5	50%/45%	ELISA
Inoue et al., 2009 [12]	Case-Control	Japan	8	12/11	50,200 (8450)	40,270 (5300)	Cases: 34.7 ± 4.9 Controls: 31.5 ± 4.7	Cases: 21.3 ± 2.3 Controls: 20.3 ± 4	-	DOT plot
Meena et al., 2018 [22]	Case-Control	India	5	75/75	35,481 (1905)	33,884 (1513)	-	-	-	ELISA
Nanda et al., 2013 [19]	Cohort	UK	7	60/240	12,768.33 (3469.7)	12,798.333 (3584.4)	Cases: 32.3 ± 7.5 Controls: 32 ± 6.41	Cases: 27.7 ± 7.2 Controls: 23.9 ± 3.35	65%/42.5%	ELISA
Seol et al., 2011 [17]	Case-Control	Korea	8	16/16	171,066.67 (18,694.8)	200,533.33 (12,029.8)	Cases: 31.9 ± 4.8 Controls: 30.8 ± 5.4	Cases: 23.7 ± 3.4 Controls: 21.6 ± 2	-	ELISA
Shangguan et al., 2009 [13]	Case-Control	China	7	26/27	4.27 (3.13)	2.65 (1.16)	Cases: 27.5 ± 1.4 Controls: 29 ± 1.5	Cases: 27.68 ± 3.28 Controls: 25.75 ± 3.03	100%/100%	ELISA
Stepan et al., 2009 [18]	Case-Control	Germany	8	16/20	24,500 (7600)	22,300 (5500)	Cases: 31.9 ± 5.9 Controls: 27.7 ± 6	Cases: 22.4 ± 3.6 Controls: 20.7 ± 2.6	-	ELISA
Vaisbuch et al., 2010 [14]	Cross-sectional	USA	9	104/134	9760.7 (2971.4)	9035.56 (2590.6)	Cases: 24.16 ± 6.74 Controls: 25 ± 6.01	Cases: 26.7 ± 4.88 Controls: 27.3 ± 7.86	62.5%/27.6%	ELISA
Wang et al., 2018 [15]	Case-Control	China	8	18/25	164.5 (44.07)	120.25 (30.79)	Cases: 28.85 ± 4.72 Controls: 31.18 ± 4.05	Cases: 21.38 ± 3.2 Controls: 22.16 ± 3.15	-	ELISA
Yliniemi et al., 2014 [16]	Case-Control	Finland	9	65/742	57,733.3 (11,599.8)	52,400 (12,032.7)	Cases: 29.3 ± 5.3 Controls: 29.8 ± 5.86	-	-	Immunoassay
Zhang et al., 2017 [23]	Case-Control	China	6	86/33	18,390 (5990)	16,000 (7470)	Cases: 30.28 ± 5.58 Controls: 30.49 ± 4.8	Cases: 22.48 ± 3.02 Controls: 21.35 ± 2.13	-	ELISA

ELISA: Enzyme-linked Immunosorbent Assay; NOS: Newcastle–Ottawa Scale; RBP4: Retinol-binding protein 4; KSA: Kingdom of Saudi Arabia; UK: United Kingdom; USA: United States of America.

**Table 2 nutrients-14-05201-t002:** Subgroup analysis of the association between retinol-binding protein 4 and preeclampsia.

Sub-Group	No. Included Studies	No. Preeclampsia	No. Normal Controls	SMD (95% CI)	Higgin’s Index *I*^2^
**Trimester of RBP4** **measurements**					
3rd trimester	10	430	418	**0.70 (0.081; 1.331)**	**90%**
1st trimester	3	139	993	0.14 (−0.264; 0.531)	72%
**Disease severity**					
Severe-preeclampsia	4	181	811	**0.71 (0.270; 1.145)**	**61%**
All spectrum	9	388	600	0.44 (−0.271; 1.143)	92%
**Studies Continent**					
Europe & America	6	286	1174	0.44 (−0.144; 1.025)	82%
Asia	7	283	237	0.64 (−0.180, 1.466)	90%
**RBP4 assay**					
ELISA	11	492	658	0.50 (−0.08; 1.08)	85%
Other methods	2	77	753	0.78 (−0.07; 1.63)	15%

Bold indicates a significant difference in the sub-group.

**Table 3 nutrients-14-05201-t003:** Meta-regression analysis for the factors that could affect the SMD of RBP4 in association with preeclampsia.

Covariate	Estimate Coefficient	Standard Error	*p*-Value	95% CI
**Study sample size**	−0.0002	0.0019	0.9001	(−0.0034; 0.0039)
**Measuring RBP4 in the 3rd trimester**	0.8831	0.9235	0.339	(−0.927; 2.6932)
**NOS**	−0.5838	0.3516	0.0969	(−1.2729; 0.1054)
**Disease severity: Severe-preeclampsia**	1.2851	0.9464	0.1745	(−0.5698; 3.1401)
**Year of publication**	−0.0575	0.1219	0.6373	(−0.2964; 0.1815)
**Continent: Europe and America**	1.0822	0.8609	0.2087	(−0.6051; 2.7696)
**Study design: Non-case-control design**	−0.0102	0.8851	0.9908	(−1.745; 1.7246)

## Data Availability

The datasets used and/or analyzed during the current study are available within the paper.

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
