# Peer review of "Association between Retinol-Binding Protein 4 Levels and Preeclampsia: A Systematic Review and Meta-Analysis"

_nutrients, 2022, doi:10.3390/nu14245201_

Round 1

Reviewer 1 Report

This article showed Retinol-binding protein 4 was significantly higher in women with preeclampsia to compared with normal controls by conducting a systematic review and meta-analysis.

This paper seems to be generally written clearly, methods are sufficient and suitable to substantiate authors’ claims. Results are reasonable. The claims are fully supported by the data and are not oversold. Discussion is based by previous literature and their results, and the limitation is described.

But if this paper should be changed in some points, it would get better. Authors should change the following major point and minor points.

Major point

The reason why the concentration of retinol-binding protein 4 varies greatly among studies was not described. In particular, the values of Seol et al., 2011, Shangguan et al., 2009, and Wang et al., 2018 were significantly different, so it is questionable whether they should be included in this meta-analysis if the reason is not known. The author should explain the reason why the values differ depending on the study and the validity of the analysis when including the study with greatly different values.

Minor points

1.     Page 1, in Abstract; the Authors should describe that heterogeneity was high in abstract.

2.     Page 4, in Figure 1; the Authors should indicate within the figure legends what the asterisks (* and **) mean in figure and should describe in figure what was the reason for the 20 exclusions on the screening (e.g. articles presenting adipokines other than Retinol-binding protein 4).

Author Response

Response to reviewers

Dear Academic Editor and respected reviewers, thank you very much for your valuable comments and insightful suggestions that we felt it has improved our manuscript.

We have highlighted the changes we made in green color and have responded to your comments as follows:

Reviewer 1

Comment

  1. The reason why the concentration of retinol-binding protein 4 varies greatly among studies was not described. In particular, the values of Seol et al., 2011, Shangguan et al., 2009, and Wang et al., 2018 were significantly different, so it is questionable whether they should be included in this meta-analysis if the reason is not known. The author should explain the reason why the values differ depending on the study and the validity of the analysis when including the study with greatly different values.

Response

The authors much appreciate your positive and helping feedback. Yes, we agreed that some of the studies have a RB4 concentration that is not homogenous with the other studies. However, as standardized mean difference (SMD) calculation is based on calculating the difference of RBP4 conc. between cases and controls within a single study. Then combining these differences by a way to expressed it as SMD. The readings of RBP4 within a single study between cases and controls is reasonable and not distorted even in the mentioned three studies. Therefore, the readings of these studies will not affect the overall pooled SMD in this study. This has been mentioned in the statistical analysis section page# 3 line#113-116.

On the other hand, according to sensitivity analysis, exclusions of the studies one by one from the final model showed neither significant changes in the overall pooled SMD of RBP4 nor in the p-value (all the readings within the calculated 95% Confidence interval), please check table S3.

Based on these points we have included these studies in the final model of the random-effect analysis of the meta-analysis.

Comment

  1. Page 1, in Abstract; the Authors should describe that heterogeneity was high in abstract.

Response

Yes, we agreed and describe the reported heterogeneity level as high. Please check the abstract page#1 line#25.

Comment

  1. Page 4, in Figure 1; the Authors should indicate within the figure legends what the asterisks (* and **) mean in figure and should describe in figure what was the reason for the 20
    exclusions on the screening (e.g. articles presenting adipokines other than Retinol-binding protein 4).

 Response

Yes, we confirmed that those asterisks are not intended and now deleted in the corrected versions. Please check figure1 page#4.

Reviewer 2 Report

Comments and Suggestions for Authors

The manuscript is a systematic review that brings together information about retinol-binding protein 4 (RPB4) in preeclampsia, but also a meta-analysis concluding that higher levels of RBP4 are associated with the risk of developing preeclampsia.

The manuscript is well written in terms of the English language, and it is also logically structured. The topic is of great interest to researchers, but also for clinicians. 

However, the fact that the meta-analysis is based on a high heterogeneity of the included studies and the authors did not find any individual study from the included ones that changed the heterogeneity measures, may raise questions about the conclusions of the study. As such, please include in the manuscript the explanation for the following major issues:

1.      To what extent the outcome of the meta-analysis could be influenced by the fact that 5 of the 13 included studies have less than 20 patients with preeclampsia and if these studies have statistical relevance in themselves?

2.      From a total of 13 studies included, 10 studies reported high levels of RBP4 in preeclampsia, 2 studies (ref. 12 and 21) low levels, and 1 almost the same values (ref 19). The 2 studies reporting low levels have included only 16 and 14 patients with preeclampsia, respectively. Please comment on how these aspects could influence the results of the meta-analysis conclusions.

3.      There are 2 studies (ref. 13 and 18) included in which the magnitude of the RBP4 concentrations are totally divergent from the other 11 studies. In that 2 studies, the average concentrations of RPB4 were of the order of units or hundreds ng/ml, and for the other 11 studies, there is a homogeneity of concentration values of the order of thousands/tens of thousands ng/ml. Please comment on how these 2 studies may influence the outcome of the meta-analysis.

4.      In Table 3 are included factors that could affect the SMD of RBP4 in association with preeclampsia. Please explain why “year of publication” is a factor and how could this factor influence the SMD of RPB4.

5.      It is well known that nulliparity is a predisposing factor for preeclampsia, along with obesity or gestational diabetes mellitus (GDM). In your meta-analysis, it was auspicious that patients with GDM were excluded, and obesity was followed via body mass index. But why was nulliparity not included in Table 1 and further in the stratified meta-analysis? Please comment on how the nulliparity may influence the outcome of the meta-analysis and why was not included?

6.      In rows 196-197, you have stated that “preeclampsia and GDM….Perhaps they have a similar pathogenesis” due to their association with insulin resistance. This is a bold statement and, therefore either you develop an explanation of the pathogenic mechanism underlying it, or simply remove it.

After the amendment of the above comments in the manuscript, I would be in favor of publishing this systematic review and meta-analysis.

Author Response

Response to reviewers

Dear Academic Editor and respected reviewers, thank you very much for your valuable comments and insightful suggestions that we felt it has improved our manuscript.

We have highlighted the changes we made in green color and have responded to your comments as follows:

Reviewer 2

Comment

  1. To what extent the outcome of the meta-analysis could be influenced by the fact that 5 of the 13 included studies have less than 20 patients with preeclampsia and if these studies have statistical relevance in themselves?

Response

Firstly, the authors much appreciated your positive feedback. Yes, we hypothesized that sample size may be a factor that affects the SMD of RBP-4. Therefore, we have included sample size in the meta-regression analysis model. However, it shows no significant effects on the SMD levels of RBP4.   This has been mentioned please see table 3 page#10 and result section page#8 line#181-183.

Comment

  1. From a total of 13 studies included, 10 studies reported high levels of RBP4 in preeclampsia, 2 studies (ref. 12 and 21) low levels, and 1 almost the same values (ref 19). The 2 studies reporting low levels have included only 16 and 14 patients with preeclampsia, respectively. Please comment on how these aspects could influence the results of the meta-analysis conclusions.

Response

Yes, we agreed that inconsistency of the reported results in previous studies allows us to conduct this systematic review and meta-analysis. Also, the effect of the sample size on the RBP4 is not significant according to the meta-regression analysis. We have added this to the limitation points as well. Please check the limitation section at the discussion page# 12 line#270.

Comment

  1. There are 2 studies (ref. 13 and 18) included in which the magnitude of the RBP4 concentrations are totally divergent from the other 11 studies. In that 2 studies, the average concentrations of RPB4 were of the order of units or hundreds ng/ml, and for the other 11 studies, there is a homogeneity of concentration values of the order of thousands/tens of thousands ng/ml. Please comment on how these 2 studies may influence the outcome of the meta-analysis.

Response

Yes, we agreed that these two studies values is not homogenous when compared to other studies. Yet, standardized mean difference is calculated based on the difference of a single study. The levels of RBP4 is homogenous between cases and controls within the same study. Therefore, the differences of RBP4 values between these two studies and other studies will not affect the overall pooled SMD of RBP4. This has been mentioned in the statistical analysis section please see page#3 line#113-116.

Comment

  1. In Table 3 are included factors that could affect the SMD of RBP4 in association with preeclampsia. Please explain why “year of publication” is a factor and how could this factor influence the SMD of RPB4.

Response

Yes, we have noticed that the SMD is increased with the year of publication. This is can be attributed to the increased sensitivity of the ELISA used over years. However, this is not significant therefore we didn’t comment on it. Also, we attempt to remove it from the model, yet the new model is not affected therefore we keep it in the final model of meta-regression analysis.  This has been mentioned please see result section page#8 line#181 – 185.

Comment

  1. It is well known that nulliparity is a predisposing factor for preeclampsia, along with obesity or gestational diabetes mellitus (GDM). In your meta-analysis, it was auspicious that patients with GDM were excluded, and obesity was followed via body mass index. But why was nulliparity not included in Table 1 and further in the stratified meta-analysis?

Please comment on how the nulliparity may influence the outcome of the meta-analysis and why was not included?

Response

Yes, we agreed and added the nulliparity data to the table1 as suggested. Also, we wrote some discussion regarding the association of preeclampsia with nulliparity, please check the discussion section page# 11 line# 247-258.

 However, only 4 studies have included nulliparity data in their manuscripts and only one study presented the results of RBP4 among nulliparous participants among cases and controls (Shangguan et al., 2009). Therefore, it is inapplicable to perform stratified meta-analysis based on "parity". Yet, we addressed this point also in the study limitations. Please check the limitation in the discussion section page#12 line#271-274.

Comment

  1. In rows 196-197, you have stated that “preeclampsia and GDM….Perhaps they have a
    similar pathogenesis
    ” due to their association with insulin resistance. This is a bold statement and,
    therefore either you develop an explanation of the pathogenic mechanism underlying it, or simply remove it.

Response

Yes, we agreed and replace this sentence with two sentences and cited two very recent references " Recently, RBP4 has been identified as a leading factor for endothelial dysfunction a landmark pathology observed in preeclampsia as well as in GDM [37,38]. Perhaps, RBp4 is the key modulator for the pathogenesis of both preeclampsia and GDM.". Please check the discussion section page# 11 line#213-215.

Round 2

Reviewer 1 Report

This paper was precisely revised, it got better. But this article should be described about the following point.

I agree the difference in mean for all studies is standardized by the pooled SD differences. However, the authors should describe the reason why the concentration of retinol-binding protein 4 varies greatly among studies, if they know. If they don't know, they should state that as well.

Author Response

Response to reviewers

Dear Academic Editor and respected reviewers, thank you very much for your valuable comments and insightful suggestions that we felt it has improved our manuscript.

We have highlighted the changes we made in green color and have responded to your comments as follows:

Reviewer 1

Comment

  1. I agree the difference in mean for all studies is standardized by the pooled SD differences. However, the authors should describe the reason why the concentration of retinol-binding protein 4 varies greatly among studies, if they know. If they don't know, they should state that as well.

Response

The authors much appreciate your positive and helping feedback. Yes, we agreed and mentioned possible causes for variations in RBP4 in these studies as the following " In this study, we observed that the reported levels of RBP4 are widely different between the included studies. This can be attributed to the differences in genetic factors that can affect the expression of RBP4 and hence its levels [48]. Moreover, degree of obesity is a documented determining factor for the levels of RBP4 [7]. Furthermore, different assay methods were used in the included studies to assess the levels of RBP4 such as ELISA, immunoassay and Dot-plot. There is a notable variation in the sensitivity of these different methods [49]. And even the ELISA, which is the widely used method has a different sensitivity based on the affinity of the antibody to RBP4 [49]. Taken together, these factors can contribute to the inter-study variations of the levels of RBP4 observed in this study ". Please check the discussion section page#  12 line#267-275.

Reviewer 2 Report

I believe that the authors have made the necessary changes and the article can be published.

Author Response

Many thanks for your positive response. 

No points were raised to respond to.